# Adenosine Methylation Level of miR-125a-5p Promotes Anti-PD-1 Therapy Escape through the Regulation of IGSF11/VSIG3 Expression

**DOI:** 10.3390/cancers15123188

**Published:** 2023-06-14

**Authors:** Gwenola Bougras-Cartron, Arulraj Nadaradjane, Marie-Pierre Joalland, Lisenn Lalier-Bretaudeau, Judith Raimbourg, Pierre-François Cartron

**Affiliations:** 1CRCI2NA, INSERM, Université de Nantes, 44035 Nantes, France; 2Institut de Cancérologie de l’Ouest, 44805 Saint-Herblain, France; 3SIRIC ILIAD, 44000 Nantes, France

**Keywords:** epitranscriptomics, methylation, adenosine, PD-1 therapy, lung cancer

## Abstract

**Simple Summary:**

Anti-PD1 therapy appears as one of the most promising anticancer therapy of recent years. Almost all patients will develop resistance over time. In this context, we have sought to identify resistant patients, in order to be able to propose a therapeutic solution. Our study has identified a circulating biomarker discriminating patients at risk of escaping PD1 therapy. This biomarker is the adenosine methylated form of the EVs/exosomal miR-125a-5p. To go further, IGSF11- and METTL3-based therapies have demonstrated in vitro efficiency, suggesting that these therapies could be used in anti-PD1 therapy-treated patients having high levels of EVs/exosomal adenosine methylated miR 125a-5p.

**Abstract:**

Background: Despite encouraging anti-tumour activity in lung cancer, anti-PD-1 therapy has encountered increasing resistance to treatment. Several companion diagnostic assays have been performed to identify patients who may benefit from this immunotherapy and to adapt this therapy in case of acquired resistance. Methods: A large panel of methods was used for the analysis of expression and methylation levels of miRNAs (qPCR, MemiRIP, …), protein/miRNA interactions (CLIP, oligo pull-down, …), and protein–protein interactions (CoIP) in cells and/or blood samples. Results: Our work highlights that the saturation of PD-1 by anti-PD1 therapies induces an immune escape phenomenon due to the overexpression of IGSF11 following adenosine methylation of miR-125a-5p. Mechanistically, we identify METTL3/KHDRBS3 and HuR as two crucial players in the methylation and the loss of the repressive function of this miRNA. Finally, our work shows that the adenosine methylation of miR-125a-5p is analyzable from EVs/exosomes from longitudinal blood samples and that such EVs/exosomes modulate the IGSF11/VSIG3 expression in lung cancer cells to promote an immune escape phenomenon. Conclusions: Our data provide a biomarker (m6A-miR-125a-5p level) and two therapeutic solutions (anti-IGSF11 antibody and METTL3 inhibitor) that could potentially address the anti-PD1 therapy failure in the context of precision and personalized medicine.

## 1. Introduction

Immunotherapy with antibodies disrupting the PD-1/PD-L1 axis provides an important breakthrough in the treatment of cancer, as these antibodies have improved the survival outcomes of patients in many cancer types [1]. For this reason, the use of antibodies against PD1/PDL1 alone or in combination with other drugs became a gold standard of first-line treatment in different advanced/metastatic tumor types. Furthermore, in addition to the growing FDA-approved indications (e.g., more than ten), a large number of clinical trials continue to investigate the efficiency of anti-PD-1 therapy alone or in combination with another anticancer therapy in various indications.

Prior to being involved in the regulation of the immune response and considered one of the most important negative immune checkpoints, PD-1 (Programmed cell Death Protein 1, UniProtKB reference: Q15116) was initially described as a gene associated with apoptosis during thymic T cell selection. Mechanistically, the canonical PD-1 pathway in T cells includes its binding with its ligands CD274/PD-L1 and CD273/PD-L2, the activation of the SHP1/2 signaling pathways, and inhibition of PI3K/AKT, MAPK, and mTOR pathways to deliver immune escape messages: suppression of T cell proliferation, cytokine production, and cytotoxic functions [2]. Consequently, the blockade of the PD-1/PD-L1 or PD-L2 axis by antibodies such as Nivolumab (Opdivo^®^), Pembrolizumab (Keytruda^®^), or Cemiplimab (Libtayo^®^) has emerged as a promising strategy to restore the anti-tumor immune response. However, some patients are completely refractory to anti-PD-1 therapy, and a large part of patients do not present durable remission.

To identify the molecular mechanisms responsible for these phenomena, most studies have focused on immune cells since PD-1 was initially described as being mainly expressed by these cells. Thus, reduced dendritic cells maturation, suboptimal T-cell activation, impaired T-cell trafficking and infiltration, stroma-dependent exclusion, additive negative immune checkpoint on immune cells, and presence of immunosuppressive cells have been identified as causes of anti-PD-1 therapy escape [3].

In addition, several authors highlighted that the PD-1 blockade on the surface of tumor cells could also play a role in the resistance to anti-PD-1 therapy. Kleffel et al. (2015) [4], Li et al. (2017) [5], Du et al. (2018) [6], Pu et al. (2019) [7], and Wang et al. (2020) [8] reported that melanoma, hepatocellular carcinoma, pancreatic ductal adenocarcinoma, and lung cancer cells expressed PD-1, respectively. As reported by Yao et al. (2018) [9], several databases (TCGA, Human protein Atlas or Cancer Cell Line Encyclopedia (CCLE)) indicate that a large variety of cancer cells express PD-1. Based on these reports, it seems legitimate to question the effect of anti-PD-1 therapy on PD-1-expressing tumor cells. In the context of lung cancer, the literature reports that the PD-1 silencing or blockade promoted cell proliferation or colony formation in the cellular assay and tumor growth in in vivo assay (Du et al. (2018) [6] and Wang et al. (2020) [8]). Thus, the use of anti-PD-1 therapy appears paradoxical: on the one hand, anti-PD-1 therapy can provide an anti-tumor response by the immune system activation, and on the other hand, anti-PD-1 therapy can provide a pro-tumor response by increasing the proliferation of tumor cells.

Extracellular vesicles (EVs, i.e., exosomes and microvesicles) and their contents (miRNA, protein, mRNA, …) have been considered as candidate biomarkers for the diagnosis, the monitoring of disease progression and the prediction of response to cancer therapy [10]. Numerous studies show correlations between the proteome and transcriptome of tumor cells and that of EVs circulating in the biological fluids (blood, urine, saliva, …) of patients with different cancers. Moreover, EVs are also considered actors of cell–cell communication via the transfer of the biological material they contain. Since Valadi et al. (2007) [11] and the description that exosomes transfer mRNA and miRNA between mast cells, a large number of articles have reinforced the major role played by EVs and particularly exosomes in intercellular communication. Our laboratory also contributed to this by demonstrating that radiotherapy-induced overexpression of exosomal miRNA-378a-3p in cancer cells limits the cytotoxicity of natural killer cells [12] and by reporting that anti-PD1 therapy induces the release of lymphocyte-derived exosomal miRNA-4315, thereby inhibiting Bim-mediated apoptosis of tumor cells [13]. Moreover, in these two articles, it appears that the longitudinal study of the expression levels of exomiR-4315 and exomiR-378 during the administration of anticancer treatments can be used as a biomarker.

In recent years, epitranscriptomic studies of miRNAs (i.e., the study of reversible chemical modifications occurring within microRNAs (whether mature or not)) are booming with the development of sequencing methods and/or the adaptation of DNA methylation analysis methods. Indeed, the work of Berulava et al. (2015) [14] with a method coupling m6A-IP and sequencing, Konno et al. (2019) [15] with the detection of methylated miRNA using MALDI-TOF-MS, Pandolfini et al. (2019) [16] with a method coupling m7G-IP and sequencing, Cheray et al. (2020) [17] with BS-miRNA-seq and the use of a method coupling m5C-IP and qPCR-Array, and Carissimi et al. (2021) [18] with the use of BS-miRNA-seq and MAmBA methods have increased the knowledge in the field of epitranscriptomics of miRNA. Moreover, some of these works have highlighted that methylated miRNAs can hold a function of biomarker. For example, Konno et al. (2019) [15] report that the adenosine methylation of miR-17-5p is a biomarker to distinguish early pancreatic cancer patients from healthy patients. Cheray et al. (2020) [17] report that the high level of cytosine methylation of miR-181a-5p was associated with poor survival of GBM patients. Briand et al. (2020) [19] report that the adenosine methylation of miR-200b-3p was associated with a prognosis of survival for GBM patients. Zhang et al. (2021) report that the level of serum circulating m6A miRNAs can be used as a biomarker for cancer detection and the development of non-invasive therapeutic monitoring [20].

To further understand the molecular mechanisms involved in the effect of anti-PD-1 therapy on lung cancer cells, we propose here an original study aiming at defining whether an epitranscriptomic reprogramming of miRNAs (i.e., their adenosine, cytosine, and guanosine methylation, here) in lung cancer cells could account for the escape of the anti-PD-1 therapy.

## 2. Materials and Methods

### 2.1. Cell Culture

A549 (ATCC#CCL-185), H1975 (ATCC#CRL-5908), H358 (ATCC#CRL-5807), and H1650 (ATCC#CRL-5883) cells were cultured in a RPMI medium supplemented with 10% of fetal bovine serum, 1% penicillin–streptomycin, and 2 mM L-glutamine. All cells were cultivated in a 5% CO_2_ incubator at a temperature of 37 °C.

### 2.2. Cytometry Analysis

Directly-labeled monoclonal antibodies were used for flow cytometry APC-mouse IGg1K isotype control (#555751, BD Biosciences, Grenoble, France) and APC-mouse anti-human CD279 (#558694, BD Biosciences, Grenoble, France). Two hundred thousand cells of each cell line were labeled at room temperature in PBS/1% BSA, washed twice in PBS, and analyzed on a BD ACCURI C6 cytometer (Paris, France). Cancer lines were gated according to their forward-scatter and side-scatter properties and excluding debris and doublets.

### 2.3. PD-1 Receptor Occupancy (Saturation) Analysis

PD-1 receptor occupancy (saturation) by Nivolumab was investigated using the in-cell ELISA method. Tumor cells (4000 cells/well) were plated in 96 wells plates for 24 h. Nivolumab (#A1307, Biovision/CliniScience, Nanterre, France) was next incubated at the indicated concentration for 3 h at room temperature. Cells were then washed four times with 1× phosphate-buffered saline (PBS). Cells were fixed using an 8% paraformaldehyde solution (15 min, room temperature). After four extensive washes, anti-PD-1-HRP (0.5 μg/mL, #10377-MM07-H, Interchim, Montluçon, France) was incubated for 2 h at room temperature, and the signal was detected using Step Ultra TMB-ELISA substrate solution according to the manufacturer’s instructions (#34028, Thermo Fisher Scientific, Illkirch-Graffenstaden, France). The saturation percentage was calculated relative to the experiment performed with a control antibody (#DDXCH0P-100, Novus, Nantes, France).

### 2.4. miRNA Extraction

miRNA extractions were performed using miRNeasy mini kit (#217004, Qiagen, Paris, France) on a QIAcube instrument according to the manufacturer’s instructions (Qiagen, Paris, France).

### 2.5. qPCR of miRNA

For miRNA expression analysis and detection from products of RNA immunoprecipitation (RIP) performed with anti-m6A antibody, miRNA was reverse transcribed using a miScript II RT kit (#218161, Qiagen, Paris, France) and analyzed by qPCR with the miScript SYBR Green PCR kit (#218076, Qiagen, Paris, France) using the specific miScript primer assays (#218300, Qiagen, Paris, France) according to the manufacturer’s instructions. Rotor-Gene Q was used as a real-time thermocycler (Qiagen, Paris, France).

### 2.6. mRNA Extraction and qPCR 

RNA extractions were performed using the RNeasy kit (#74116, Qiagen, France) and QIAcube according to the manufacturer’s instructions (Qiagen, Paris, France). Reverse transcription was then performed with a QuantiTect Reverse Transcription Kit (#205311, Qiagen, Paris, France), and PCR was performed with a QuantiFast SYBR Green PCR Kit (#204156, Qiagen, Paris, France) and Rotor-Gene Q (Qiagen, Paris, France). Primers used were the QuantiTect Primers assay (#249900, Qiagen, Paris, France) and, more precisely, RPLP0 (QT00075012), KHDRBS3 (QT00065296), METTL3 (QT00036540), and HuR (QT00037856).

### 2.7. Methylation Level of miRNAs

For RNA immunoprecipitation (RIP), two rounds using 5 μg of anti-m6A (#ab208577, Abcam, Paris, France), anti-m7G (#6655, Biovison, Milpitas, CA, USA), and anti-m5C (#61255, Active Motif, Nantes, France) antibodies and 5 μg of small RNA were performed. The reaction was carried out using a Dynabeads protein G IP kit with some modifications (#10007D, Thermo Fisher Scientific, Paris, France) such as described by Berulava et al. (2015) [14] and Briand et al. (2021) [19]. As a control, IP was performed using IgG (#ab18443, Abcam, Paris, France) instead of an anti-m6A antibody. miRNAs obtained from RIP were reverse transcribed using a miScript II RT kit (#218161, Qiagen, Paris, France) and analyzed using the miScript miRNA PCR array human cancer pathway kit (#331221, Qiagen, Paris, France) according to the manufacturer’s instructions. Fold enrichment was next calculated using the Ct value obtained from qRT-PCR performed with input miR, IP-IgG, IP-m6A, and the 2^−ΔΔCt^ formula.

### 2.8. siRNA Transfection

siRNA directed against METTL3, KHDRBS3, and HuR (#sc-92172, #sc-40922, and sc-35619 Santa Cruz, Morzine, France) were used in this article. siRNA-A (#sc-37007, Santa Cruz, Morzine, France), i.e., a scrambled sequence devoid of specific degradation of any cellular message, was used as control. In a six wells culture plate, 2.10^5^ cells were incubated for 24 h at 37 °C in a CO_2_ incubator. Then, 60 pmol of siRNA mixture (prepared with siRNA Transfection reagent, #sc-29528, Santa Cruz, France) was incubated on cells for 7 h at 37 °C in a CO_2_ incubator. Without removing the siRNA mixture, we next added 1 mL of normal growth medium containing two times the normal serum and antibiotics concentration for 24 h. Then, cells were expanded for 48 h in a normal culture medium. Thus, analyses were realized about 72 h after the siRNA incubation.

### 2.9. Immunoprecipitation

After two cold PBS 1× washes (10 mL), three 100 mm dish cultures at confluence 75% (2.10^5^ cells) were incubated with 500 μL of lysis buffer (20 mM TrisHcl pH 7.4, 150 mM de NaCl, 1 mM EDTA, 1% Triton + Protease cocktail inhibitor) for 5 min on ice. Then, cells were scraped and centrifuged in a 1.5 mL Eppendorf tube (4 °C, 15 min, 14,000× *g*). After a transfer in a new tube, the supernatant is used for protein dosage (#23225, Pierce BCA Protein Assay Kit, Thermo Scientific, Paris, France). At this step, the supernatant can be stored at −80 °C. Then, we performed of pre-clean step. For that, 20 μL of magnetic beads (#10004D, DynaBeads, Thermo Scientific, Paris, France) were washed twice with 1 mL of lysis buffer in a 1.5 mL Eppendorf tube. After the last wash, 500 μg of lysate was added to magnetic beads for 2 h at 4 °C on a revolver tube mixer (Labnet International, Issy-les-Moulineaux, France) to saturate aspecific interactions. After magnetic separation, the lysate is then collected to be incubated (overnight, 4 °C, revolver tube mixer) with the antibody (4 μg of anti-METTL3 (#ab195352, Abcam, France) or IgG as control (#2729S, Cell Signaling/Ozyme, Paris, France). Then, the lysate-antibody mixture was incubated (2 h, 4 °C, revolver tube mixer) with 20 μL of saturated magnetic beads (as previously described). Four washes were next performed (two with lysis buffer and two lysis buffers devoid of triton). Finally, beads were resuspended in 100 µL of Buffer NH_4_HCO_3_ at 50 mM pH = 8, and a supernatant (containing immunoprecipitation products) was stored for future use.

### 2.10. miRNA Pull-Down Assay

Cellular extracts were performed by incubating 10 min on ice 2.10^6^ cells in cell lysis buffer (85 nM KCl, 0.5% NP40, 5 mM HEPES, pH 7.4, three volumes of buffer for one cell volume). After incubation, the mixture was sonicated in a Bioruptor/Diagenode (15 cycles, high, 30 s on/off). After centrifugation (10 min, 14,000× *g*, 4 °C), the supernatant was transferred to a fresh tube to be quantified (Qubit Method, Thermo Fisher, Paris, France).

Streptavidin M280 Dynabeads (#11205D, Thermo Fisher, Paris, France) were washed three times in binding/washing buffer (10 mM Tris-HCl, pH 7.5, 1 mM EDTA, 2 M NaCl), twice with 50 μL of 0.1 NaOH/0.05 M NaCl buffer, once with 50 μL of 0.1 M NaCl. Beads were finally resuspended in 50 μL of 0.1 M NaCl and transferred in a new DNA low-binding tube. Then, 800 μg of protein was incubated with 50 μL of beads for 15 min at room temperature.

Then, the tube was placed on a magnetic stand for 2–3 min, and the supernatant was collected for future use.

In parallel, 500 pmol of biotinylated mimic miRNA was then incubated with beads for 15 min at room temperature under gentle rotation. Then, the tube was placed on a magnetic stand for 2–3 min to collect miRNA-immobilized beads. After two washes in binding/washing buffer, miRNA-immobilized beads were resuspended in 100 μL of Protein binding buffer (20 mM Hepes (pH 7.3), 50 mM KCL, 10% glycerol, and 5 mM MgCl_2_. Add the following reagents before use: 1/1000 vol. of 1 M Dithiothreitol (DTT), Phosphatase inhibitor (100×), Proteinase inhibitor (100×), and RNase Inhibitor (25~1000 U/mL, for RNA application).

The supernatant was then added, and the mixture was incubated for 30 min at room temperature.

The tube was next placed on a magnetic stand for 1 min, and the supernatant was discarded. After three washes, elution was performed using 20 μL of 4× Laemmli buffer (#1610747, BioRad, Marnes-la-Coquette, France). After incubation for 10 min at 95 °C, samples were loaded onto an SDS-PAGE gel, and proteins were detected by western blotting.

### 2.11. Western Blot

After a denaturation step (dry-bath/95 °C/10 min), samples are placed in the wells of a 4–15% Mini-PROTEAN electrophoresis gel (#4568084, Bio-Rad, Marnes-la-Coquette, France), and the electrophoresis (90 V) is performed in the presence of Tris/Glycine/SDS Electrophoresis Buffer (#1610732EDU, Bio-Rad, Marnes-la-Coquette, France). A transfer (30 V, 90 min RT) is performed on a PVDF membrane (Trans-Blot Turbo Mini PVDF, #1704156, Bio-Rad, Marnes-la-Coquette, France) in the presence of a cold Tris/Glycine transfer buffer (#1610734EDU, BioRad, Marnes-la-Coquette, France). The membrane is then saturated with the saturation solution (5% milk, PBS 1×). Primary and secondary antibodies (METTL3#MA5-27527, Thermo Fischer, Paris, France; KHDRBS3#ab68515, Abcam, Paris, France, RBMX# PA5-99433, Thermo Fischer, Paris, France; HuR# 39-0600, Thermo Fischer, Paris, France; HRP-anti-Rabbit IgG (#111-035-006, Jackson Immunology, West Grove, PA, USA) and HRP-anti-Mouse IgG (#115-036-75, Jackson Immunology)) are incubated in milk1%-PBS 1×. The HRP signal is revealed using the ChemiDoc imager (Bio-Rad, Marnes-la-Coquette, France) and the ECL-Clarity kit (#1705061, Bio-Rad, Marnes-la-Coquette, France).

### 2.12. Chromatin Immunoprecipitation (ChIP) Analyses

ChIP was performed using the ChIP-IT Express kit (Active Motif, Carlsbad, CA, USA) according to the manufacturer’s instructions. The cross-linking step was performed by treating the cells with 37% formaldehyde solution for 15 min at room temperature. Sonication was performed with the Bioruptor Plus (8 cycles 30 s ON/90 s OFF) (Diagenode, France). The QuantiFast SYBR Green PCR Kit and Rotor-Gene Q (Qiagen, Paris, France) were used to perform the qPCR. Antibodies used were: Anti-IgG (#ab2410, Abcam, Paris, France) and anti-ELK1 (sc365876x, Santa Cruz, Morzine, France). The primers used here are gagagcatttgctgggttg and ttgtcatcccgacttgctcct.

### 2.13. Cell Cytotoxicity Assay

This assay permits estimating the lysis of desired target cells (H1975 here) via an effector immune cell of choice (PBMC activated by IL2, here). In this assay, the lysis of H1975 cells is determined by labeling these cells with fluorescent molecules, co-incubating them with IL2-activated PBMC, then measuring the release of the labeled molecule in the supernatant. DELFIA EuTDA Cell Cytotoxicity (#AD0116, Perkin Elmer, Villebon-sur-Yvette, France) reagents were used as previously described (Briand et al. 2019) [21]. For PBMC isolation, whole blood was diluted with an equal amount of 1× phosphate-buffered saline (PBS) and placed on top of Ficoll-Paque PLUS (#17-1440-02, VWR, Rosny-sous-Bois, France) in tubes for centrifugation at room temperature for 20 min at 1000× *g*. PBMCs were next carefully collected from the interface layer between the blood plasma and Ficoll solution. The collected PBMCs were twice washed with 1× PBS, then cultured in an RPMI1640 medium containing 10% FBS, 0.1 nM human IL-2 (Diaclone Research, Besançon, France), and 55 μM β-mercaptoethanol for 24 h. BATDA-labeled target cells and PBMC effector cells (effector to target cell ratio: 10:1) were incubated (2 h, 37 °C, 5% CO_2_) together to allow redirected lysis. The supernatant containing TDA released by dead cells was incubated with Europium solution to allow the stable formation of a highly fluorescent Europium-TDA chelate (EuTDA). The amount of EuTDA was quantified by analysis of the relative fluorescence intensity (FL_sample_) at 520 nm. Spontaneous fluorescence intensity was determined with target cells without effector cells. The fluorescence intensity of maximal lysis (FL_max_) was determined using target cells and DELFIA^®^ lysis buffer. The percentage of specific cell lysis was calculated as 100 × (FL_sample_ − FL_Spon_)/(FL_max_ − FL_Spon_).

### 2.14. RT-qPCR Analysis

RNA extract is performed using RNeasy Mini QIAcube Kit and QIAcube (#1038703, Qiagen, Paris, France). RT-qPCRs are performed using the QuantiTect Reverse Transcription Kit (#205313, Qiagen, Paris, France), QuantiFast SYBR Green PCR Kit (#204057, Qiagen, France), QuantiTect Primer Assays (#249900, Qiagen, Paris, France), and Rotor-Gene Q as a real-time thermocycler (Qiagen, Paris, France). Reference gene RPLP0 was used with the 2^−∆∆Ct^ relative quantification method.

### 2.15. CLIP

CLIP assays were performed using a RIP-assay kit (#RN1005 CliniScience, Nanterre, France) from 10 million per sample of UV-crosslinked cells (150 mJ/cm^2^ of UVA (Bio-link, Vilber, Marne-la-Vallée, France)) according to the manufacturer’s instructions. IP assays were performed in the presence of 15 μg of anti-GW182 (#RNP033P, CliniScience, Nanterre, France) overnight at 4 °C.

### 2.16. Isolating Exosomal miRNA from Blood

From the blood sample collected in K + EDTA tubes, 4–5 mL of blood was centrifugated for 10 min/1900× *g*/4 °C. Supernatants were carefully transferred into a new centrifuge tube and centrifuged a second time at 16,000× *g* for 10 min at 4 °C to remove cellular debris. Plasmas were aliquoted in 2 mL tubes and were frozen at −80 °C until use. A total of 1 mL of plasma was processed for the isolation of miRNA using the ExomiRNeasy Midi kit (#77144, Qiagen, Paris, France) according to the manufacturer’s instructions.

### 2.17. Exosome Isolation

As described in our previous articles [12,13], ExoQuick kits (#EXOTC10A1 and #EXOCG50A1, Ozyme, Saint-Cyr-l’École, France) were used according to the manufacturer’s instructions. Exosome total protein concentration was determined using the Bradford assay (Bio-Rad Laboratories, Marnes-la-Coquette, France), and exosomes were stored at −80 °C until use. As already mentioned, the Nanosight experiment was performed to validate the use of the “exosome” term due to the isolation of EVs with a size predominantly between 80 and 120 nm, i.e., values commonly characterizing exosomes. Consequently, the term “exosome” is used in this article.

### 2.18. Patients Data

Plasma was collected from patients treated at the “Institut de Cancérologie de l’Ouest” (ICO, http://www.ico-cancer.fr, accessed on 4 May 2023). All patients recruited gave signed informed consent. All the samples collected and the associated clinical information were registered in the database (N° DC-2018-3321) and validated by the French research ministry. Biological resources were stored at the “Centre de Ressources Biologiques-Tumorothèque (CRB)” (Institut de Cancérologie de l’Ouest, Saint-Herblain, France) [22]. All clinical and radiologic data were collected from the electronic medical records stored at the Institut de Cancérologie de l’Ouest.

### 2.19. Statistical Analysis and Results

Except when indicated, data are representative of the mean and standard deviation calculated from three independent experiments. The significance of the differences in means ± standard deviations was calculated using the Student *t*-test. *p* < 0.05 was used as a criterion for statistical significance.

## 3. Results

### 3.1. Nivolumab Saturation of PD-1 in H1975 Lung Cancer Cells Affects the Methylation Status of miRNAs and the Cellular Immunogenicity

To identify a cellular model on which to study the impact of anti-PD-1 therapy on miRNA methylation levels, we analyzed PD-1 expression on the surface of a panel of four lung cancer cell lines (A549, H1975, HT358, and HT1650). Flow cytometry analysis showed that all lung cancer cell lines expressed PD-1 on their surface and that the H1975 cell line was the cell line expressing the most PD-1 on its surface (Figure 1A). We thus retained this cell line to carry out the continuation of our experiments. In parallel, we then arbitrarily chose to work with Nivolumab, an anti-PD-1 very frequently used in the treatment of lung cancer (more than 290 clinical trials listed on the ClinicalTrails.gov website with the “lung cancer and Nivolumab” keywords). According to our saturation experimental design (Figure 1B), we noted that 0.1 μg/mL Nivolumab saturated 80% of PD-1 expressed on the surface of H1975 lung cancer cells (H1975^NivoSat^) (Figure 1C). We thus retained this concentration to carry out the continuation of our experiments.

Changes in miRNA expression levels were analyzed using a qPCR miR array, and changes in adenosine (m6A), cytosine (m5C), and guanosine (m7G) methylation levels were analyzed using RNA immunoprecipitation with an anti-methyl-base-antibody followed by qPCR miR array (n = 84). Among all the miRNA measured by the miR Array, the expression fold change between H1975 and H1975^NivoSat^ cells appeared homogeneous on the whole (fold change varies within the range −5/+5) (Figure 1D, horizontal axis). By contrast, when considering only methylated miRNA (Figure 1D, vertical axis), a few of them could be distinguished from the group since they increased out of the range. A gain of adenosine methylation (m5A) was observed in 5 miRNAs in H1975^NivoSat^: hsa-miR-18a-5p, hsa-miR-19a3-p, hsa-miR-100-5p, hsa-miR-125a-5p hsa-miR-146b-5p (Figure 1D, left). A gain of cytosine methylation (m5C) was observed in 7 miRNAs in H1975^NivoSat^: hsa-miR-27a-3p, hsa-miR-126-3p, hsa-miR-132-3p, hsa-miR-181c-5p, hsa-miR-184, hsa-miR-200c-3p, and hsa-miR-210-3p (Figure 1D, middle). Such specific modification of guanosine methylated miRNA could not be identified in H1975^NivoSat^ cells (Figure 1D, right).

The TAM2.0 bioinformatic tool was next used to determine the functional implications of miRNAs having a gain of cytosine and adenosine methylation [23]. Thus, we observed that miRNAs having a gain of cytosine and adenosine methylation were mainly associated with the immune response function (Figure 1E). In other words, this analysis suggests that the immune response towards H1975^NivoSat^ cells could be modified. To investigate this hypothesis, we performed a cell cytotoxicity assay in which the H1975 and H1975^NivoSat^ lysis is performed by IL2-activated PBMC. Our data indicated that the IL-2-activated PBMC-induced lysis of H1975^NivoSat^ cells was lower than the one of H1975 cells (Figure 1F).

### 3.2. Adenosine Methylation of miR-125a-5p Regulates Its Repressive Function toward VSIG3/IGSF11 and Influences the Cellular Immunogenicity

We then extended our study by investigating how variations in cytosine and adenosine methylation of miRNAs could modulate the immunogenicity of H1975^NivoSat^ cells. For this purpose, we hypothesized that the previously observed gain of miRNAs methylation could affect the expression of immune response actors. Among a panel of 8 immune actors B7H3/CD276 (uniprot: Q5ZPR3), B7H4/VTCN1 (uniprot: Q7Z7D3), GAL9/HOM-HD-21 (uniprot: O00182), TNFRSF14/CD270 (uniprot: Q92956), PD-1/CD279 (uniprot: uniprot: Q15116), PD-L1/CD274 (uniprot: Q9NZQ7), PD-L2/CD273 (uniprot: Q9BQ51) and IGSF11/VSIG3 (uniprot: Q5DX21), RT-qPCR indicated that only *IGSF11* expression was increased in H1975^NivoSat^ in comparison with H1975 (Figure 2A).

Among the 156 miRNAs identified by the miRDB website as regulators of IGSF11 (Appendix A), only two miRNAs, miR-125a-5p and miR-181c-5p, were also included in the list of 12 miRNAs having a gain of adenosine/cytosine methylation (Figure 2B).

Based on this finding, we next analyzed whether the gain of adenosine and cytosine methylation of miR-125a-5p and miR181c-5p could be responsible for the increase in IGSF11 expression. For this purpose, two sets of experiments were performed.

First, we analyzed the ability of unmethylated and methylated miR-125a-5p and miR181c-5p to regulate the *IGSF11* expression. For that, we designed unmethylated and methylated mimic miRs in which the positions of adenosine and cytosine methylation were predicted by the presence of DRCAH and CG sequences. Interestingly, mutation of adenosine at position 9 and 17 and cytosine at position 15 in miR-125a-5p and miR-181c-5p abolished the METTL3/METTL14 and DNMT3A/AGO4-mediated methylation of these miRs (Appendix A). Thus, it was observed that neither the mimic-miR181c-5p transfection nor its cytosine-methylated form altered *IGSF11* expression (Appendix A). On the contrary, the transfection of the unmethylated mimic-miR-125a-5p down-regulated the *IGSF11* expression, while a similar quantity of adenosine-methylated mimic-miR-125a-5p did not alter *IGSF11* expression (Figure 2C). Under these conditions, we also observed that the quantity of miR-125a-5p co-immunoprecipitated with GW182 remained unchanged in experiments performed with adenosine-methylated mimic-miR-125a-5p in comparison with a control condition, while the quantity of miR-125a-5p co-immunoprecipitated with GW182 strongly increased in experiments performed with unmethylated mimic-miR-125a-5p (Figure 2C). The results of this first set of experiments support the idea that miR-125a-5p down-regulates the IGSF11 expression, while the adenosine-methylated form of this miR loses this ability.

Secondly, we analyzed the Sinefungin (SFG) effect, an inhibitor of METTL3, the main actor responsible for adenosine methylation of miRNA, on the adenosine methylation of miR-125a-5p and the IGSF11 expression. Thus, we observed that SFG treatment (i) decreased the level of m6a-miR-125a-5p without changing the expression level of miR-125a-5p, (ii) restored the miR-125a-5p enrichment on GW182 that had been decreased by Nivolumab saturation, and (iii) decreased the *IGSF11* and IGSF11 expressions (Figure 2D). The fact that Nivolumab increased the adenosine methylation of miR-125a-5p and the IGSF11 expression was also observed in two other models based on the consideration of H1650 and A549 cells (Appendix A). This second set of data supports the idea that the adenosine methylation level of miR-125a regulates the IGSF11 expression.

Finally, we observed that SFG and anti-IGSF11 have the ability to restore an IL-2-activated PBMC-induced lysis cell of H1975^NivoSat^ to a level comparable to the one observed in H1975 cells (Figure 2E). In other terms, this last result reinforces the fact that IGSF11 plays a crucial role in the low level of IL-2-activated PBMC-induced lysis of H1975^NivoSat^.

### 3.3. METTL3/KHDRBS3 Promotes the Adenosine Methylation of miR-125a-5p

To explain the molecular mechanism involved in the gain of adenosine methylation of miR-125a-5p, we first investigated a putative change in METTL3 or FTO expression since these proteins are the main enzymes governing the adenosine methylation level of miRNA. ELISA indicated that METTL3 and FTO expression was the same in H1975 and H1975^NivoSat^ cells (Figure 3A).

We next hypothesized that the gain of adenosine methylation of miR-125a-5p seen in H1975^NivoSat^ cells could be due to a process that we called RNA binding protein-directed miRNA methylation. In this process, the miRNA methylation could be due to the formation of a complex including METTL3 and an RNA binding protein (RBP) having the ability to bind miR-125a-5p in a sequence-specific manner. To investigate this hypothesis, we first in silico searched for an RBP with the ability to bind miR-125a-5p. The database of RNA-binding protein (http://rbpdb.ccbr.utoronto.ca/index.php, accessed on 4 May 2023) identified six RBP (QKI, ACO1, YBX1, RBMX/HNRNPG, KHDRBS3, and ELAVL1/HuR) likely to bind miR-125a-5p (Appendix A). By integrating this list with the list of “METTL3 interactors” (according to BioGrid), only the KHDRBS3 protein appears in common (Appendix A). Based on this analysis, we then hypothesized that METTL3 could form a complex with KHDRBS3 to adenosine–methylate the miR-125a-5p. IP experiments indicated that KHDRBS3 was co-immunoprecipitated with METTL3 only in H1975^NivoSat^ cells (Figure 3B and Appendix A).

We next performed siRNA-mediated METTL3 and KHDRBS3 down-regulation in order to question if METTL3 and KHDRBS3 cooperate in methylating miR-125a-5p. Thus, we observed that siRNA-directed against KHDRBS3 decreased the adenosine methylation of miR-125a-5p without affecting its expression level (Figure 3C). Under these conditions, the expression and adenosine methylation level of miR-411-5p (a miRNA that includes the DRACH sequence of adenosine methylation but not the KHDRBS3 binding sequence) (Appendix A) remained unchanged (Figure 3C).

We also noted that siRNA-directed against METTL3 decreased the adenosine methylation of both miR-125a-5p and miR-411-5p without modifying their expression levels (Figure 3C). These last data support the idea that METTL3 and KHDRBS3 form a complex responsible for the adenosine methylation of miR-125a-5p.

It was previously reported that cells treated with Nivolumab have a higher activated ERK level than the cells treated with a control antibody [8], ERK activating the transcriptional activity of ELK1 [24] and KHDRBS3 is an ELK1-regulated gene (according to the Harmonizome website, accessed on 4 May 2023) (Appendix A). To explain the METTL3/KHDRBS3 co-immunoprecipitation seen in H1975^NivoSat^ cells and not in H1975 cells, we thus hypothesized that this could be due to the KHDRBS3 overexpression following the activation of the Nivolumab/PD-1/ERK/ELK1 signaling. To incriminate this pathway in our cells, experiments were performed in the presence of ERK inhibitor (Ravoxertinib, #HY-15947, MedChem Express, Princeton, NJ, USA) and/or siRNA directed against ELK1. As expected, chemical and biological inhibition of ELK1 decreased the KHDRBS3 expression (Figure 3D). The direct involvement of ELK1 on the KHDRBS3 expression was here supported by the fact that ELK1 enrichment on the *KHDRBS3* promoter was more important in H1975^NivoSat^ than in H1975 and decreased in the presence of the ELK1 inhibitor or siRNA (Figure 3D). Thus, these results together support the idea that the presence of the METTL3/KHDRBS3 complex in H1975^NivoSat^ cells is due to the activation of Nivolumab/PD-1/ERK/ELK1 signaling.

### 3.4. HuR Blocks the Recruitment of m6A-miR-125a-5p to GW182

To explain why the adenosine–methylated form of miR-125a-5p (m6a-miR-125a-5p) is not recruited by GW182, we hypothesized that this could be due to steric blockade induced by the recruitment of an “adenosine-methylated-binding protein (m6A-binding protein)”. Based on the literature, we established a list of 35 m6A-binding proteins (Appendix A). By integrating this list with the list of RBP-binding miR-125a-5p, we identified two proteins that we suspected to play the role of a steric blocker: HuR and RBMX (Figure 4A and Appendix A). In order to determine whether these proteins have the ability to bind m6A-miR-125a-5p, a pull-down experiment was performed. Western blot performed from this experiment shows that RBMX remains in the flow through when subjected to pull-down with a biotinylated miR-125a-3p mimic attached to avidin–sepharose beads, while HuR physically interacts with the biotinylated miR-125a-3p mimic (Figure 4B and Appendix A). We then completed our study by analyzing the impact of siRNA-induced downregulation of HuR on the level of adenosine methylation of miR-125a-5p, the recruitment of this miR by GW-182 and HuR, and the IGSF11 expression. After observing that siRNA-HuR decreased the expression level of *HuR*, our data associate this context with the maintenance of the adenosine methylation and expression levels of miR-125a-5p, the increase oinmiR-125a-5p recruitment by GW182 and the decrease in IGSF11 expression (Figure 4C). In parallel, we also observed that HuR recruited miR-125a-5p in H1975^NivoSat^ cells but not in H1975 cells (Figure 4D). Our investigations also show that treatment of H1975^NivoSat^ cells with SFG (a METTL3 inhibitor decreasing the adenosine methylation level of miR-125a-5p and increasing the recruitment of this miR by GW182 (Figure 2D)) reduced the recruitment of miR-125a-5p by HuR (Figure 4D). Thus, all these results support the idea that HuR recruited the adenosine-methylated form of miR-125a-3p to block its recruitment with GW182.

### 3.5. Investigation on the Adenosine Methylation of Exosomal miR-125a-5p in Two Lung Cancer Patients

To determine whether the adenosine methylation level of miR-125a-5p could be associated with positive or negative response of lung cancer patients treated with anti-PD-1 therapy, we questioned the adenosine methylation level of exosomal miR-125a-5p (m6A-exomiR-125a-5p).

In vitro, we observed an increase of m6A-miR-125a-5p levels in exosomes issued from H1975^NivoSat^ cells without observing changes in the expression level of exomiR-125a-5p (Figure 5A). We also addressed this question for the two other antibodies used in patients for anti-PD1 therapy used in the clinic: Pembrolizumab (#A1306, Biovision/CliniScience, Nanterre, France) and Cemiplimab (#A2249, Biovision/CliniScience, Nanterre, France). Our data indicated that the three anti-PD1 therapies tested increased the adenosine methylation of exomiR-125a-5p. Under this condition, we also observed an increase of IGSF11 expression (Appendix A).

Based on this finding, we analyzed the m6A-exomiR-125a-5p and exomiR-125a-5p levels from longitudinal blood samples from two lung cancer patients treated with anti-PD1 therapy.

Patient#A was a 69-year-old male, former smoker, treated for a lung adenocarcinoma TTF1+ without EGFR, KRAS, or BRAF mutation or ALK or ROS1 translocation but an expression of 100% of PDL1 on the tumor cell. This patient showed synchronous bone, lung, and brain metastases. He received pembrolizumab in the first line setting from 24 April 2018 to 11 June 2020 (35 cycles) and presented, as the best response, a partial response (−42%) according to RECIST 1.1. obtained at the first evaluation (8 weeks). The patient is always on follow-up and presents until now no disease progression. Our analyses showed a rapid decrease of m6A-exomiR-125a-5p levels without changes in the expression level of exomiR-125a-5p (Figure 5B). Patient#B was a 53-year-old male, current smoker, with a lung adenocarcinoma TTF1+ PDL1 0% without EGFR, KRAS, or BRAF mutation or ALK and ROS1 translocation but STK11 mutation. The tumor was initially locally advanced (IIIC). The patient received first-line carboplatine pemetrexed bevacizumab from August 2018 to November 2018 with a partial response allowing the realization of closing mediastinal radiotherapy. Three months after the end of radiotherapy, the CT scan showed the apparition of bilateral lung metastases and multiple brain metastasis. A second line with Nivolumab was started on June 2019. The first evaluation after four cycles showed a progression according to RECIST 1.1 criteria confirmed on a second CT Scan one month later. Nivolumab was stopped after six cycles. Our analyses showed a rapid increase of m6A-exomiR-125a-5p levels without changes in the expression level of exomiR-125a-5p (Figure 5C). The report of these two cases provides proof of concept that the anti-PD-1 therapy-mediated modulation of the adenosine methylation level of miR-125a-5p can be analyzed by considering exomiR-125a-5p in longitudinal blood samples of lung cancer patients.

Finally, we analyzed the effect of patient-derived exosomes on the IGSF11 expression of H1975 cells.

For patient#A, exosomes from D63 and D105 blood samples decreased IGFS11 expression in H1975 cells more strongly than exosomes from D2 blood samples (Figure 5D). The ability of patient#A exosomes to induce the decrease of IGSF11 expression in H1975 cells is even higher as it contains low adenosine methylated exomiR-125a-5p. The fact that transfection of D105 exosomes with anti-miR-125a-5p results in the loss of repression of IGSF11 reinforces the idea that miR-125a-5p was the exosomal actor of this repression.

For patient#B, exosomes from D42 and D80 blood samples increased IGFS11 expression in H1975 cells compared with exosomes from D0 blood samples (Figure 5D). In other words, a high amount of adenosine–methylated exomiR-125a-5p in exosomes inhibits their ability to trigger IGSF11 repression in H1975 cells. The dual fact that transfection of miR-125a-5p into D80 exosomes decreases IGSF11 expression while transfection of miR-125a-5p-m6A does not confirm the role played by adenosine methylation of miR-125a-5p in its repressive function towards IGSF11. Thus, these experiments performed with patient-derived exosomes reinforce the interest in considering the adenosine methylation of miR-125a-5p on the miR-125a-5p-mediated regulation of IGSF11 expression.

## 4. Discussion

PD-1 signaling pathway characterization in cancer and immune cells has provided meaningful progress in understanding the resistance to PD-1 therapy but also gave rise to a paradox. Indeed, blockade, invalidation, or inhibition of PD-1 in melanoma (Kleffel et al.) [4] and HCC (Li et al. [5]) reduced tumor growth, while the blockade of PD-1 in lung and colon cancer cells promoted tumor growth (Du et al. [6], Wang et al. [8]). In addition, these articles underline the fact that the PI3K/AKT, MAPK/ERK, and/or mTOR signaling pathways play a crucial role, via their activation or inhibition, in the pro- or anti-tumor role played by the PD-1 inhibition/blockade. Our work adds a new pathway of resistance to PD-1 therapy by identifying the METTL3-mediated adenosine methylation of miR-125a-5p and the regulation of IGSF11 expression as the two central points of the PBMC-induced lysis escape phenomenon.

The involvement of the METTL3-mediated adenosine methylation of miR-125a-5p and the IGSF11 expression in the PBMC-induced lysis escape phenomenon provides two distinct strategies to limit the acquisition of this phenomenon. Indeed, our work supports the idea that the PBMC-induced lysis escape phenomenon observed in H1975^NivoSat^ can be abrogated using either anti-IGSF11 antibody-based immunotherapy or targeted therapy based on inhibition of adenosine methylation of miR-125a-5p. In addition to SFG, we also observed that UZH1a, a potent and selective METTL3 inhibitor [25], and siRNA directed against METTL3 reduced the adenosine methylation level of miR-125a-5p and decreased the IGSF11 expression in H1975^NivoSat^ cells (Appendix A). Our work is not the only one to focus on these two therapeutic areas, which are IGSF11 antibody-based immunotherapy and METTL3 inhibitor-based therapy. Several studies support the idea that IGSF11 is a promising new target for the immunotherapy of gastrointestinal and hepatocellular carcinomas (Watanebe et al. [26]) and glioma (Ghouzlani et al. [27]). In this regard, iOmx therapeutics claims to have an anti-IGSF11 antibody in development (https://iomx.com/pipeline, accessed on 4 May 2023). The search for METTL3 inhibitors is also a growing field, especially since the recent acceleration of epitranscriptomics with the characterization of methylations of small non-coding RNAs such as miRNAs [14,16,17,28]. Concerning METTL3, the main actor of the adenosine methylation of miRNAs, several inhibitors have been described, such as UZH1a [25] and others [29,30]). In this regard, STORM Therapeutics develops STM2457, a small-molecule inhibiting METTL3 presenting a potential therapeutic interest in the fight against leukemia [31] (https://www.stormtherapeutics.com/science/pipeline/, accessed on 4 May 2023).

By identifying the overexpression of the negative immune checkpoint IGSF11 as a player in the loss of PBMC-induced lysis, our work echoes the idea that resistance to anti-PD-1 therapy may result from the overexpression of other negative immune checkpoints. In this respect, the work of Koyama et al. [32] shows that resistance to anti-PD-1 therapy may be due to the overexpression of the negative immune checkpoint TIM-3. In addition, by showing that anti-IGSF11 increased the PBMC-mediated lysis of H1975^NivoSat^, our work contributes to the idea that combining several immunotherapies may be more efficient than using single immunotherapy and/or could limit the apparition of PD-1 therapy resistance [33].

In many cases, resistance to anticancer treatment is favored by changes in DNA methylation patterns. By identifying a gain of cytosine and adenosine methylation in seven and five miRNAs, respectively, our results show that changes in miRNA methylation patterns are also a molecular event associated with resistance to anticancer treatment. The role of METTL3 and adenosine RNA methylation in response to anti-PD-1 therapy has already been demonstrated in the literature. Indeed, Wang et al. [8] have recently shown that adenosine methylation of the 3’UTR regions of STAT1 and IRF1 transcripts influences the response to anti-PD-1 therapy.

At the level of fundamental knowledge of miRNA epitranscriptomic regulation mechanisms, this paper introduces two new concepts. Indeed, through the demonstration of the role played by METTL3/KHDRBS3 in the adenosine methylation of miR-125a-5p, our work introduces, for the first time, the notion of RBP-directed miRNA methylation, i.e., a mechanism by which an RBP (here KHDRBS3) addresses a miRNA methyltransferase (here METTL3) on a miRNA to methylate it. Interestingly, this mechanism is reminiscent of the TF-directed DNA methylation mechanism in which a transcription factor (TF) addresses a DNMT to DNA to methylate it.

Through the demonstration of the role played by HuR in blocking the recruitment of m6A-miR-125a-5p by GW182, our work introduces, for the first time, the notion of a blocker of miR-mediated gene silencing via the blockage of adenosine–methylated miRNA recruitment by GW182. Interestingly, this mechanism is reminiscent of the mechanism of transcription blocking via the recruitment of methylated-binding proteins (MeCP2, Kaiso, …) on methylated DNA [34,35,36].

Finally, our data technically provide a first proof of concept that the adenosine methylation level of miR-125a-5p can be studied using miRNA encapsulated in circulating blood exosomes. Of course, the prognosis character of the evolution of the adenosine methylation of exomiR-125a-5p will require evaluation in large cohorts of patients treated with anti-PD-1 therapy. However, based on the promising results obtained in the two patients reported here, the study of the evolution of the adenosine methylation of exomiR-125a-5p during anti-PD-1 therapy represents an attractive alternative approach to select the right time when the right patient is eligible for personalized medicine combining anti-PD-1 therapy with other immunotherapies (such as the one targeting IGSF11) or therapy targeting METTL3. Moreover, the methylation level of miR-125a-5p could be used as an early marker of response to anti-PD-1 therapy and thus serve to guide the realization of other more invasive examinations (and/or more costly in technical and human resources) to evaluate the response to anti-PD-1 therapy. In other terms, our article suggests that the analysis of the methylation level of exomiR-125a-5p is part of a context of precision and personalized medicine: “precision” in terms of the use of therapy targeting IGSF11 or METTL3 and “personalized” in terms of the patient via the possible detection of the right time to treat the right patient with the right therapy.

Our paper differs from the one of Konno et al. (2019) [15] in that we are talking here about a biomarker of response to therapy and not a diagnostic biomarker. Moreover, our work relates to an exosomal miRNA and not to a tissue miRNA (such as in Konno et al. [15]). Interestingly, the present study complements the study of Guyon et al. (2020) [13], showing that exomiR-4315 from lymphocytes exposed to anti-PD-1 therapy is a source of resistance in tumor cells receiving this exomiR via regulation of the pro-apoptotic protein Bim. Indeed, these two studies highlight the existence of two exomiRs (exomiR-125a-5p and exomiR-4315) having different cellular origins (tumor cells and lymphocytes), different mechanisms of action (inhibition of apoptosis and immune escape) but both being potential biomarkers of resistance to anti-PD-1 therapy. This point is a current line of research in our laboratory via the constitution of a prospective cohort of patients with lung cancer receiving anti-PD-1.

## 5. Conclusions

In conclusion, this study identifies the miR-125a-5p/IGSF11 as an axis involved in the resistance to anti-PD-1 therapy and the adenosine methylation of this miRNA as a crucial mechanism regulating this axis. We also show that the adenosine methylation of exosomal miR-125a-5p could be a potential biomarker of anti-PD-1 therapy failure in lung cancer patients. However, this last result needs to be confirmed in a large cohort of lung cancer patients. In addition, our studies suggest that anti-IGSF11- and METTL3 inhibitor-based therapies could be used to abrogate this mechanism of anti-PD-1 therapy failure.

## Figures and Tables

**Figure 1 cancers-15-03188-f001:**
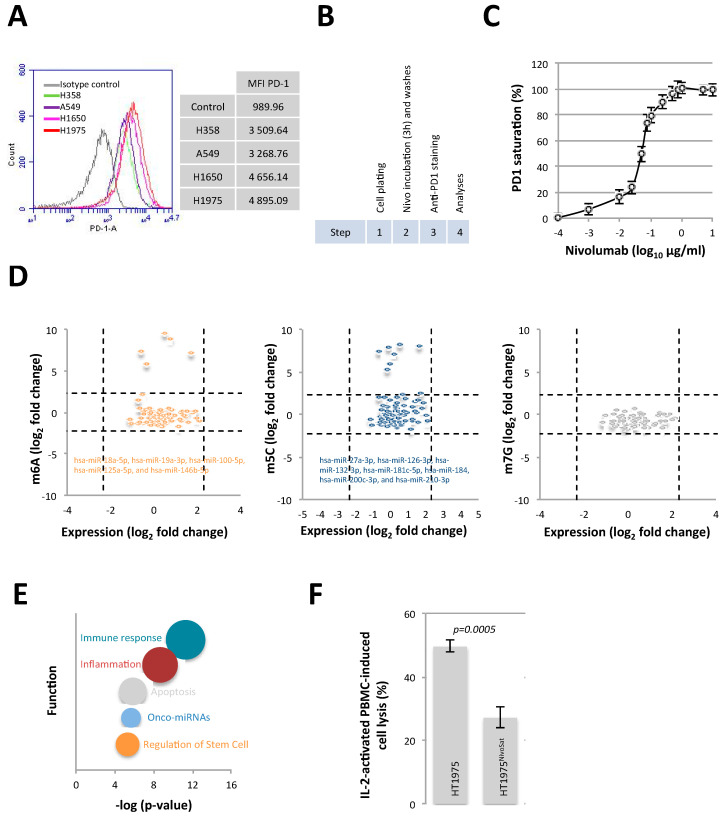
Nivolumab saturation of PD-1 of H1975 lung cancer cells affects the methylation status of miRNAs and cellular immunogenicity. (**A**) Analysis of the level of PD-1 on four different cell lines was done by flow cytometry. The grey line represents the staining with the isotype control antibody. The means of fluorescent intensity were reported in the table. (**B**) Illustration of experimental design. (**C**) Graph illustrates the percentage of PD-1 saturation by Nivolumab at the surface of H1975 cells. (**D**) Graphs illustrate the expression and the adenosine (**left**), cytosine (**middle**), and guanosine (**right**) methylation levels. For the expression and methylation variations, we used a cutoff value of 5 (or log2 = 2.32) (dashed line). The list of miRNAs with m6A and m5C variation greater than 5 is shown in the corresponding graphs. (**E**) Bubble chart illustrates the Top5 functions associated with differentially methylated miRNAs between H1975 and H1975^NivoSat^ cells. (**F**) Immunogenicity of H1975^NivoSat^ cells was compared with the one of H1975 cells through the analysis of their lysis-induced PBMC.

**Figure 2 cancers-15-03188-f002:**
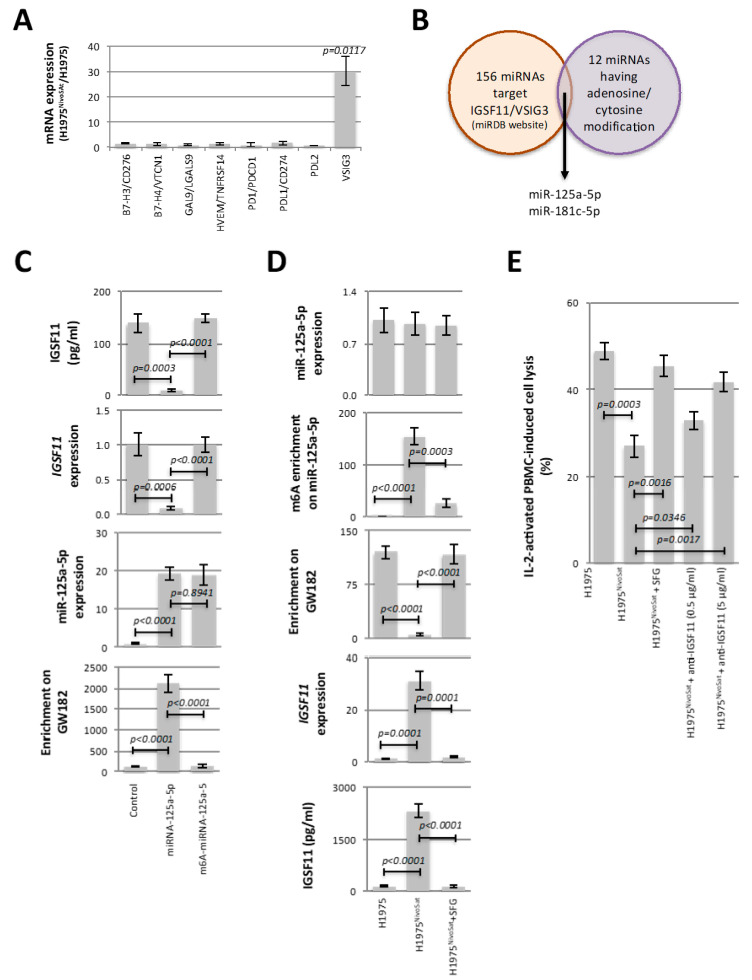
Adenosine methylation of miR-125a-5p regulates its repressive function toward VSIG3/IGSF11 and influences cellular immunogenicity. (**A**) RT-qPCR analysis of expression changes of genes involved in cell immunogenicity. (**B**) Venn diagram highlighting the existence of 2 miRNAs (miR-125a-5p and muR-181c-5p) having a modification of m6A or m5C and being described as targeting IGSF11. (**C**) Graphs illustrate the impact of transfection of wild-type and adenosine methylation forms of miR-125a-5p on IGSF11 at the protein and mRNA level and on their binding to GW182 within H1975 cells. (**D**) Graphs illustrate the impact of Sinefungin (SFG) on the expression and adenosine methylation level of miR-125a-5p, on the interaction of this miRNA with GW182, and on the protein and mRNA expression levels of IGSF11 in H1975^NivoSat^ cells. H1975 cells were used as control. (**E**) Immunogenicity of H1975^NivoSat^ cells was compared with the one of H1975^NivoSat^ cells treated with Sinefungin or anti-IGSF11 through the analysis of their PBMC-induced lysis. H1975 cells were used as control.

**Figure 3 cancers-15-03188-f003:**
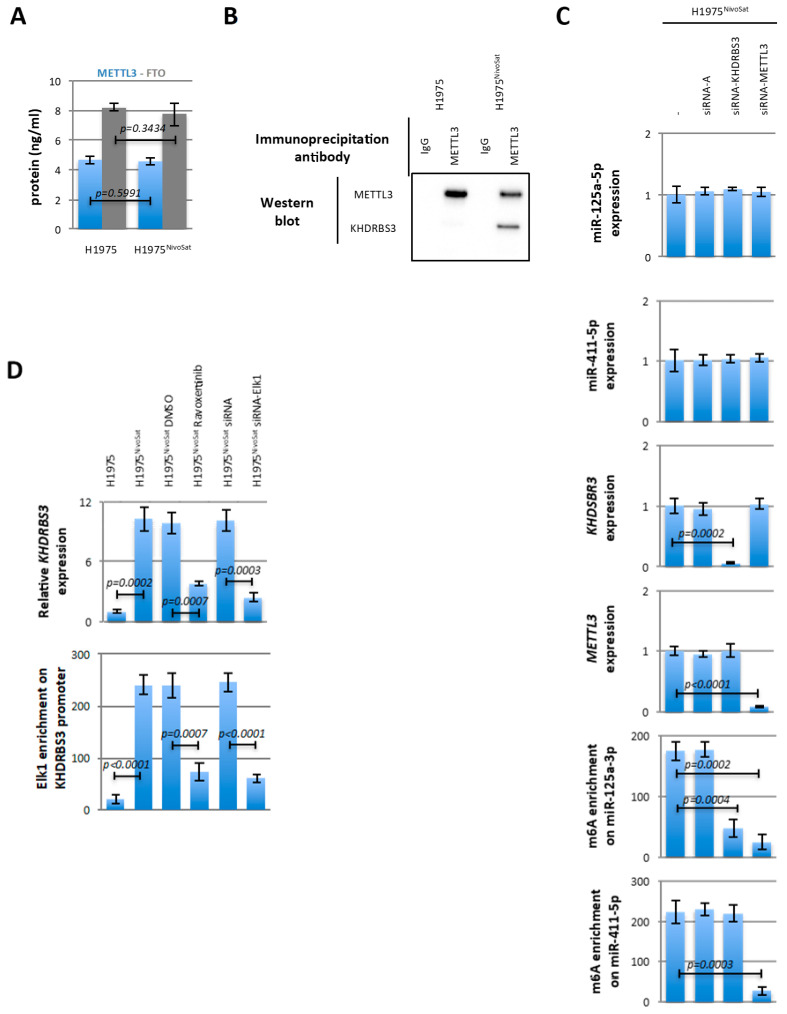
METTL3/KHDRBS3 promotes the adenosine methylation of miR-125a-5p. (**A**) Graph showing the ELISA assay of METTL3 (#ABX381419, CliniSciences, Nanterre, France) and FTO (#K4921, Biovision, Nanterre, France) expression level in H1975 and H1975^NivoSat^ cells. (**B**) The presence of METTL3 and KHDRBS3 proteins was analyzed by western blot in immunoprecipitation products made with antibodies directed against METTL3 and IgG (control) and protein extracts from H1975 and H1975^NivoSat^ cells. (**C**) The transfection impact of H1975^NivoSat^ cells with siRNA control (siRNA-A) and siRNA directed against KHDSBR3 and METTL3 on the expression level of miR-125a-5p and 411-5p (RT-qPCR), KHDRBS3 and METTL3 transcripts (RT-qPCR), the level of enrichment in m6A of miR-125a-5p and 411-5p. (**D**) Impact of chemical (Ravoxertinib, 100 nM, 72 h) and biological (siRNA) inactivation of ELK1 on the expression level of the KHDRBS3 transcript and the enrichment level of ELK1 on the KHDRBS3 promoter. DMSO and siRNA-A were used as control.

**Figure 4 cancers-15-03188-f004:**
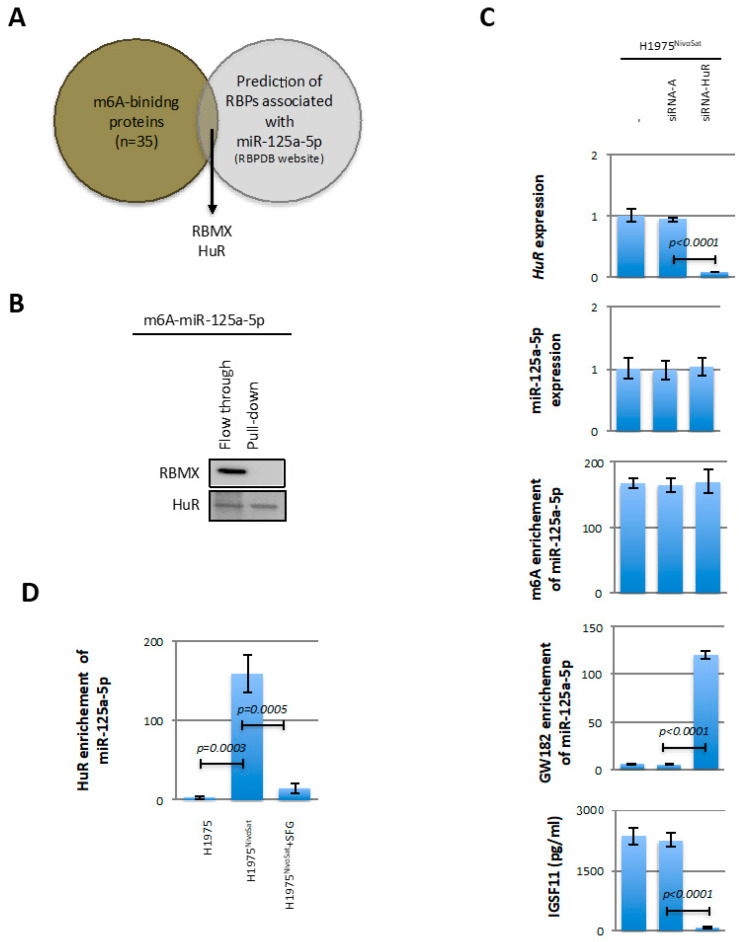
HuR blocks the recruitment of m6A-miR-125a-5p to GW182. (**A**) Venn diagram illustrating the identification of RBMX and HuR as proteins with both m6A and miR-125a-5p binding capacity. (**B**) Western blot illustrating the ability of HuR to bind to m6A-miR-125a-5p, unlike RBMX (which remains in the flow through). (**C**) Analysis of the impact of siRNA invalidation of HuR on the adenosine methylation and expression levels of miR-125a-5p, on the miR125a-5p/GW182 interaction, and on the expression level of IGSF11. (**D**) Analysis of miR-125a-5p/HuR interaction within H1975, H1975NivoSat, and METTL3 inhibitor (SFG) treated cells.

**Figure 5 cancers-15-03188-f005:**
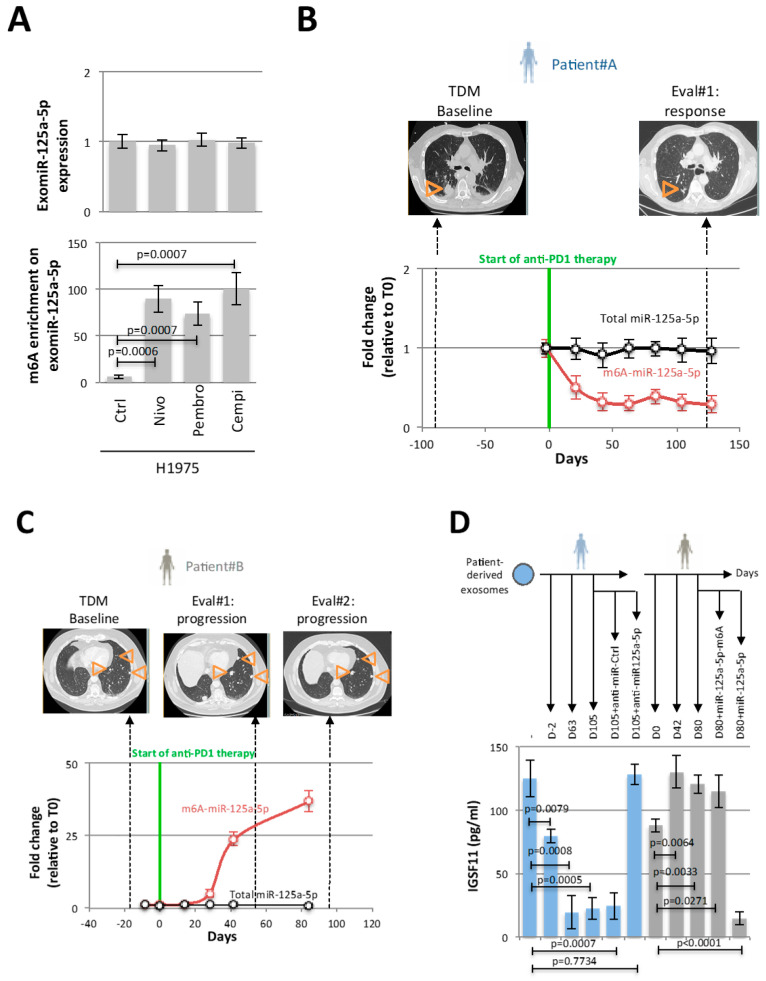
Investigation on the adenosine methylation of exosomal miR-125a-5p in two lung cancer patients. (**A**) Graphs illustrate the fact that modification of adenosine methylation of the miR-125a-5p can be analyzed using exosomal miR-125a-5p. Three anti-PD-1 therapies were used to investigate this point in H1975 cells. (**B**) The expression and adenosine methylation levels of miR-125a-5p were analyzed using exosomes from longitudinal blood samples taken from patient#A between 2 MRIs assessing the fate of his tumor. (**C**) The expression and adenosine methylation levels of miR-125a-5p were analyzed using exosomes from longitudinal blood samples taken from patient#B between 2 MRIs assessing the fate of his tumor. Orange triangles indicate the location of tumor masses (**D**) Analysis of the impact of exosomes obtained from several blood samples of patients #A and #B on the expression level of IGSF11 in H1975 cells.

## Data Availability

Raw data, extended data tables, and analyses are available upon reasonable request from the corresponding author.

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
