# Peer review of "Adenosine Methylation Level of miR-125a-5p Promotes Anti-PD-1 Therapy Escape through the Regulation of IGSF11/VSIG3 Expression"

_cancers, 2023, doi:10.3390/cancers15123188_

Round 1
Reviewer 1 Report (Previous Reviewer 2)
In this revised manuscript, the authors have done good job to address my previous questions, especially they showed the evidences of the RBP-directed miRNA methylation mechanism and how the recruitment of a m6A-binding protein (HuR) blocking the miR/GW182 interaction. There are several questions still need to be addressed.
1. The writing needs to be improved. For example,
1) The grammar is wrong in lines 442-445.
2) Gene names should be italicized, while protein names should not. For example, Elk1. For human genes or proteins, the name should be in upper case.
3) Is “IGSF11mRNA expression” same as “ IGSF11 expression”? If they both mean gene expression, “IGSF11 expression” should be used.
4) Line 112, what is RMPI medium?
5) What do “mimic-miR181c-5p” and “mimic-miR-125a-5p” mean? There are lots of “mimic” in the manuscript, please clarify the definition.
6) Line 477-478, does the “biotinylated miR-125a-3p mimic” mean m6A-miR-125a-5p? This could be more specific.
These are some examples. The authors should check the whole manuscript very carefully.
2. I am confused about the method section 2.3. Please confirm the Nivolumab treatment was performed after cells were fixed.
Some writing is confusing and needs to be improved.
Author Response
Review#1. Comments and Suggestions for Authors
In this revised manuscript, the authors have done good job to address my previous questions, especially they showed the evidences of the RBP-directed miRNA methylation mechanism and how the recruitment of a m6A-binding protein (HuR) blocking the miR/GW182 interaction. There are several questions still need to be addressed.
- The writing needs to be improved. For example,
1) The grammar is wrong in lines 442-445.
2) Gene names should be italicized, while protein names should not. For example, Elk1. For human genes or proteins, the name should be in upper case.
3) Is “IGSF11mRNA expression” same as “ IGSF11 expression”? If they both mean gene expression, “IGSF11 expression” should be used.
4) Line 112, what is RMPI medium?
5) What do “mimic-miR181c-5p” and “mimic-miR-125a-5p” mean? There are lots of “mimic” in the manuscript, please clarify the definition.
We used the term mimic miR when we used custom synthesized single stranded miRs. The mimic miRs have the same sequence as the natural miRs. We have explain this in the new version of our article.
6) Line 477-478, does the “biotinylated miR-125a-3p mimic” mean m6A-miR-125a-5p? This could be more specific.
These are some examples. The authors should check the whole manuscript very carefully.
>> We thank the reviewer for pointing out these errors which we have corrected in the new version of our article.
- I am confused about the method section 2.3. Please confirm the Nivolumab treatment was performed after cells were fixed.
>> We thank the reviewer for pointing out this error which we have corrected in the new version of our article.
Reviewer 2 Report (Previous Reviewer 1)
This article is a paper previously submitted by Bougras-Cartron et al and revised. The authors demonstrated that anti-PD-1 therapy increased the level of m6A-miR-125a-5p and regulated IGSF11 expression. This study is interesting and could provide a potential combination therapy for anti-PD-1 resistant patients. However, some minor modifications need to be made before publication.
The authors have considerably improved their work by confirming their data on 2 other cell lines.
Minor modifications.
- Fig 1 B shows 2 cycles of: 3h of nivolumab followed by washouts which is not written in the materials and methods. is this an error?
- The authors indicate that they have carried out statistical tests which do not appear on any of the figures with histograms. It would be nice to show the significance by stars.
Author Response
Review#2. Comments and Suggestions for Authors
This article is a paper previously submitted by Bougras-Cartron et al and revised. The authors demonstrated that anti-PD-1 therapy increased the level of m6A-miR-125a-5p and regulated IGSF11 expression. This study is interesting and could provide a potential combination therapy for anti-PD-1 resistant patients. However, some minor modifications need to be made before publication.
The authors have considerably improved their work by confirming their data on 2 other cell lines.
Minor modifications.
- Fig 1 B shows 2 cycles of: 3h of nivolumab followed by washouts which is not written in the materials and methods. is this an error?
>> We thank the reviewer for pointing out this error which we have corrected in the new version of our article.
- The authors indicate that they have carried out statistical tests which do not appear on any of the figures with histograms. It would be nice to show the significance by stars.
>> we thank the reviewer for this suggestion. The p-values of the t-tests have been added to the figures
This manuscript is a resubmission of an earlier submission. The following is a list of the peer review reports and author responses from that submission.
Round 1
Reviewer 1 Report
In the article "Adenosine methylation level of miR-12a-5p drives anti-PD-1 therapy....", G Bougras-Cartron et al. describe a new biomarker and 2 therapeutic solutions to explain the non response to anti-PD1.
General comment.
The authors studied only one cell line, H1975, while other cell lines also express PD1 on their surface, notably H1650, which has an almost identical expression rate. A second model is needed to confirm.
The figures are difficult to read. There is a large number of typing errors in the text, a rereading is more than necessary.
The authors mention having performed the tests with other anti-PD1 and mention additional figures. I have no record of these figures.
Specific comment.
Figure 1B two cycles of Nivolumab/wash were performed with 3h incubation of nivolumab? in the material and method it is mentioned only 1 cycle and 4h exposure? please clarify this point.
The authors must greatly improve and complete this work before it can be published
Reviewer 2 Report
In this manuscript, Bougras-Cartron et al. demonstrated that anti-PD-1 therapy increased m6A-miR-125a-5p level and regulated IGSF11 expression. The study is interesting and could provide potential combination therapy for anti-PD-1 resistant patients. However, this study only focused on one lung cell line and limited number of miRNAs and genes, it was overstated to use “drives” in the title. In addition, several issues must be addressed to strengthen the findings.
Comments:
1, Have the authors investigated the methylation sites of miR-125a-5p? How did they design the m6A-miR-125a-5p used in figures 2C and 3D?
2, Is there any explanation about how the anti-PD-1 (nivolumab) affecting miR-125a-5p methylation? Does it decrease METTL3 level or activity?
3, What is the mechanism of m6A-miR-125a-5p affecting binding by GW182? Does the methylation of other miRNAs have the same effect?
4, Figure 1D, why did the authors use the cutoff of 5? It is very stringent. There are clear trends that many miRNAs expression is upregulated at least 2 times by nivolumab treatment. Those miRNAs could Play roles in anti-PD-1 therapy.
5, The IL-2-activated PBMC-induced cell lysis should be introduced, and the purpose of this assay should be explained.